# The Acute Effects of a Single Dose of Molecular Hydrogen Supplements on Responses to Ergogenic Adjustments during High-Intensity Intermittent Exercise in Humans

**DOI:** 10.3390/nu14193974

**Published:** 2022-09-24

**Authors:** Ahad Abdulkarim D. Alharbi, Noriaki Iwamoto, Naoyuki Ebine, Satoshi Nakae, Tatsuya Hojo, Yoshiyuki Fukuoka

**Affiliations:** 1Graduate School of Health and Sports Science, Doshisha University, Kyoto 610-0396, Japan; 2Human Augmentation Research Center, National Institute of Advanced Industrial Science and Technology, Kashiwa II Campus, The University of Tokyo, Chiba 277-0882, Japan

**Keywords:** hydrogen, acid status, muscle deoxygenation, high intensity exercise, peak power, exercise performance

## Abstract

This research examined the effects of single-dose molecular hydrogen (H_2_) supplements on acid-base status and local muscle deoxygenation during rest, high-intensity intermittent training (HIIT) performance, and recovery. Ten healthy, trained subjects in a randomized, double-blind, crossover design received H_2_-rich calcium powder (HCP) (1500 mg, containing 2.544 μg of H_2_) or H_2_-depleted placebo (1500 mg) supplements 1 h pre-exercise. They performed six bouts of 7 s all-out pedaling (HIIT) at 7.5% of body weight separated by 40 s pedaling intervals, followed by a recovery period. Blood gases’ pH, PCO_2_, and HCO_3_^−^ concentrations were measured at rest. Muscle deoxygenation (deoxy[Hb + Mb]) and tissue O_2_ saturation (S_t_O_2_) were determined via time-resolved near-infrared spectroscopy in the vastus lateralis (*VL*) and rectus femoris (*RF*) muscles from rest to recovery. At rest, the HCP group had significantly higher PCO_2_ and HCO_3_^−^ concentrations and a slight tendency toward acidosis. During exercise, the first HIIT bout’s peak power was significantly higher in HCP (839  ±  112 W) vs. Placebo (816  ±  108 W, *p* = 0.001), and HCP had a notable effect on significantly increased deoxy[Hb + Mb] concentration during HIIT exercise, despite no differences in heart rate response. The HCP group showed significantly greater O_2_ extraction in *VL* and microvascular (Hb) volume in *RF* during HIIT exercise. The HIIT exercise provided significantly improved blood flow and muscle reoxygenation rates in both the *RF* and *VL* during passive recovery compared to rest in all groups. The HCP supplement might exert ergogenic effects on high-intensity exercise and prove advantageous for improving anaerobic HIIT exercise performance.

## 1. Introduction

Molecular hydrogen (H_2_) is a colorless, tasteless, odorless, flammable gas [1]. H_2_ is also a minimal molecule that can quickly diffuse through the alveoli into the blood and circulate throughout the body during breathing. It can penetrate the cellular membrane and rapidly diffuse into cell organelles, playing major biological roles. Moreover, H_2_ can easily pass through the blood–brain barrier, although most antioxidant compounds cannot [2]. H_2_ possesses a selective free radical and inflammation scavenging ability; it is thought to have no effect on physiologically reactive species (e.g., H_2_O_2_), since it can selectively reduce ·OH and ONOO^−^ [3]. Molecular hydrogen is believed to have several advantages for applications in sports science [4,5,6]. In addition, H_2_ can be supplied to the body through multiple routes of administration and can be used with minimal side effects as it is excreted by exhaling. One of the most recently applied methods for administering H_2_ is through the use of hydro-calcium powder (HCP), which uses extracted calcium to absorb and trap highly concentrated hydrogen molecules (International Patent No. 4472022).

Our previous study revealed that when 1500 mg of HCP was well absorbed in subjects’ intestinal tract for 3 consecutive days, it led to slight acidosis and to increases in the venous partial pressure of CO_2_ (PCO_2_) and bicarbonate ion (HCO_3_^−^); in addition, the HCP significantly impacted the muscle O_2_ delivery/utilization ratio in the rectus femoris (*RF*) muscles but not in the vastus lateralis (*VL*) muscles during incremental exercise, without impacting the subjects’ exercise performance [7].

Dietary HCP supplementation thus impacted the muscle spatial heterogeneity in muscle deoxygenation (deoxy[Hb + Mb]) kinetics, and, by extension, the O_2_ delivery (QO_2_-to-VO_2_ matching). Because it is possible that the effect of HCP might be more pronounced on *RF* muscle with predominant fast-twitch-fiber-recruitment [8,9,10], it was speculated that an investigation of the effects of HCP during an anaerobic high-intensity intermittent training (HIIT) exercise in which fast-twitch-fiber-recruitment is dominant might reveal more about the influence of HCP on QO_2_-to-VO_2_ matching and O_2_ extraction in these muscles.

HCP increases HCO_3_^−^ and thus might delay fatigue by increasing the extracellular buffer capacity, which might improve exercise performance [11,12,13]. Additionally, as slight increases in PCO_2_ might be associated with increased cerebral blood flow (CBF) in humans [14,15], and diminished CBF during exercise or physical activity results in a reduced motor drive to working muscles, consequently negatively affecting the tolerance to whole-body exercise and possibly performance [16,17]. Thus, the increased levels of PCO_2_ as result of HCP ingestion might further influence anaerobic exercise performance.

Thus, it was hypothesized that the administration of H_2_ might (i) impact the muscle spatial heterogeneity responses and blood gas profile and (ii) exert ergogenic effects on power output and performance during HIIT exercise. To test this hypothesis, the present study was conducted to determine whether HCP supplement causes different responses to muscle deoxy[Hb + Mb] and the responses of blood gas values at rest, during anaerobic HIIT exercise, and recovery compared to a placebo supplement.

## 2. Materials and Methods

### 2.1. Subjects

Ten healthy males with no pre-existing or contradicting medical conditions or metabolic abnormalities participated. All were active members of the same university-affiliated track and field team (mean ± SD; age 20 ± 1 years; body weight 66.9 ± 5.9 kg; and height 172.4 ± 7.2 cm). All subjects are members of college athletic team and had been training for 100 m event for ≥5 years and averaged their 100 m race-record is 11.12 ± 0.38 s. Written informed consent was obtained from all subjects after a detailed explanation of the purpose of the study, procedures, possible risks, and benefits of participation. The study conformed to the Declaration of Helsinki, and the ethical committee of Doshisha University approved all the study procedures (No. 18012).

### 2.2. Supplements

Using a crossover double-blind design, the subjects were examined twice and given a single dose of four capsules of either an H_2_-rich calcium powder (HCP) supplement (ENAGEGATE, Tokyo, Japan) or calcium powder as a placebo (ENAGEGATE) 1 h before the experiment (Figure 1A). The HCP and placebo capsules were identical, and each capsule contained approx. 375 mg of either HCP or only calcium powder for the placebo. The amount of hydrogen in each HCP capsule was 0.636 µg, providing 2.544 µg as the total dose of H_2_.

### 2.3. Experimental Overview

Following our previous study’s protocol [7], the subjects were familiarized with all measurement techniques, and a familiarization test was conducted a few weeks prior to the start of the experiments; the testing sessions were separated by a 1- to 2-week washout period. All sessions were completed in Doshisha University in a custom-made environmental chamber (model LP-2.5PH-SS, NK Systems, Osaka, Japan) maintained at a temperature of 25 °C and 50% relative humidity.

In the 24 h prior to the testing session, the consumption of alcohol and caffeine was prohibited, and if exercise or training was carried out by the subject, the intensity, the timing, and duration were matched for both sessions. The subjects were also instructed to record and replicate their dietary intake for dinner the previous night and for breakfast, which was consumed at home ≥ 6 h prior to the start of the session. The experimental procedure was carried out 3 h postprandial following a standardized meal for all subjects, which consisted of a small meal replacement bar (Calorie Mate block plain, four blocks, Otsuka Pharmaceutical, Tokyo, Japan) and one bottle of caffeine-free barley tea (Healthy Mineral Barley Tea, 600 mL, ITO EN, Tokyo, Japan).

### 2.4. Exercise Test

The subjects performed a high-intensity intermittent training (HIIT) exercise protocol using a cycle ergometer (Fujin-raijin, OCL, Tokyo, Japan). After 3 min of rest, the subject performed a 5 min warm-up exercise at a workload of 60 W. They then performed all-out pedaling for 7 s at a load of 7.5% of body weight (BWT). After 5 min of rest, the subject then performed six repetitions of the 7 s all-out pedaling at 7.5% BWT separated by 40 s intervals of pedaling at a workload of 60 W (i.e., the HIIT protocol). This protocol was adapted to repeatedly stress the phosphocreatine (PCr)/ATP system [18] and to target the recruitment of fast-twitch-dependent muscle, which is greater in short duration sprints [19]. A post-HIIT cool-down exercise was performed at a workload of 40 W for 5 min. The recovery period was maintained for 10 min in a passive resting state (Figure 1). The observed mean heart rate (HR) reached approx. 80% of the age-predicted maximum HR even during this HIIT protocol (Figure 1B).

### 2.5. Blood Sampling

A 1 mL blood sample was collected from each subject at rest through a peripheral venous catheter placed in the subject’s forearm, allowing for free movement of the elbow and hands during the experiment. The sample was then analyzed for blood gas, electrolytes, and the metabolic profile with a portable blood analysis system (epoc^®^, Siemens Healthcare, Tokyo, Japan), which provided the measurements of the acid status of pH, the partial pressure of O_2_ (PO_2_), the partial pressure of CO_2_ (PCO_2_), and the values of bicarbonate (HCO_3_^−^), base excess of the extracellular fluid (BE(ecf)), base excess in the blood (BE(b)), hemoglobin (Hgb), and hematocrit (Hct). The metabolic status of lactate (Lac), glucose (Glu), creatinine (Crea), and serum electrolytes including the sodium (Na^+^), potassium (K^+^), chloride (Cl^−^) concentrations, and the Aniongap (AGap) and Aniongap with potassium (AGapK) were also measured for all subjects.

### 2.6. Measurements

Each subject’s HR was measured by a chest-strapped Polar H10 HR monitor (Polar Electro Oy, Kempele, Finland). The absolute values of oxygenated (oxy[Hb + Mb]), deoxygenated (deoxy[Hb + Mb]), and the total hemoglobin and myoglobin concentration (total[Hb + Mb]) were sampled from the vastus lateralis (*VL*) and rectus femoris (*RF*) muscles of the subject’s right leg by a time-resolved near-infrared spectroscopy (TR-NIRS) system (C12707, Hamamatsu Photonics, Hamamatsu, Japan). The TR-NIRS optodes were kept in place using an elastic flexible black thigh sleeve that prevented light penetration to the optodes and ensured that their positions were fixed during the experiment protocol without compromising the subjects’ free movements. The TR-NIRS optodes for each subject were repositioned in the same location for all testing sessions. The TR-NIRS data were corrected for adipose tissue thickness (ATT) with the use of the Bowen et al. (2013) method [20]. The ATT was measured using a B-mode ultrasound system (Aixplorer Multiwave™, Supersonic Imagine, Aix-en-Provence, France). The tissue O_2_ saturation (S_t_O_2_) was calculated using oxy[Hb + Mb]/total[Hb + Mb]; for full details, see Alharbi et al. [7].

The TR-NIRS and HR results were calculated as the mean value for 30 s at sampling points, including rest and the recovery periods at the 1st min (Rec.1), 3rd min (Rec.3), 5th min (Rec.5), 7th min (Rec.7), and 10th min (Rec.10) (Figure 1). The peak values for the TR-NIRS deoxy[Hb + Mb], total[Hb + Mb], HR, and peak power output (PP) were determined as the highest detected value during the 7 s all-out bout for each set (Peak.1~6). The S_t_O_2_ was determined as the lowest detected value during the 7 s all-out bout.

### 2.7. Statistical Analysis

All data are expressed as the mean ± standard deviation (SD) and were analyzed using the statistical package IBM SPSS, PC program, ver. 28.0 (IBM, Tokyo, Japan). Normality was tested using the Shapiro–Wilk test. The results of these tests indicate that the data did not show evidence of non-normality. Mean values of parameters between the HCP and placebo supplement groups at rest and the mean values across the six peaks of HIIT exercise were compared using a paired *t*-test. A two-way analysis of variance (ANOVA) for repeated measurers was employed to identify the significance of differences in variables between the HCP and placebo supplement groups at the sampling points. A post hoc Bonferroni test was implemented when a significant *F*-value was noted. Probability (*p*)-values < 0.05 were considered significant. Associated metrics of effect size, Cohen’s d for paired *t*-tests (d) and partial eta squared (η^2^) within repeated measures ANOVA were calculated. Cohen’s d effect sizes can be interpreted as: ≥0.20 (small), ≥0.50 (medium), and ≥0.80 (large) and partial eta squared (η^2^) effect sizes can be interpreted as: ≥ 0.01(small), ≥0.06 (medium), ≥0.14 (large) [21]. Pearson’s correlation coefficient (r) for liner relationship correlation graphs were also calculated and can be interpreted as: ≥0.20 (weak), ≥0.40 (moderate), ≥0.70 (strong), ≥0.90 (very strong) [22,23].

## 3. Results

### 3.1. At Rest

The mean values of acid-base, TR-NIRS, and HR profiles are presented in Table 1. The blood gas status values in the HCP group showed the following significant differences from the placebo group: Lower PO_2_ (HCP: 31.3 ± 5.9 vs. placebo: 41.3 ± 8.7 mmHg, t_(6)_ = −2.860, *p* = 0.029, d = 1.081), higher PCO_2_ (55.0 ± 4.7 vs. 50.7 ± 5.0 mmHg, t_(6)_ = 3.606, *p* = 0.011, d = 1.363) and HCO_3_^−^ (30.5 ± 1.6 vs. 29.2 ± 1.7 mmol∙L^−1^, t_(6)_ = 4.156, *p* = 0.006, d = 1.571), and significantly lower SO_2_%, higher BE and Hgb (all values, *p* < 0.05). Although the pH was slightly lower in the HCP group at 7.354 ± 0.021 (placebo: 7.369 ± 0.018), the difference was not significant; however, a large effect size was present (t_(6)_ = −2.331, *p* = 0.059, d = 0.881). The metabolic profiles of Lac, Glu, and electrolytes were within the standard ranges at rest with no significant difference between the HCP and placebo groups. The at-rest TR-NIRS profiles of the *RF* and *VL* muscles and HR also showed no significant between-group difference.

### 3.2. HR Response during HIIT Protocol and Recovery

The HR response from rest to recovery and the peak values adapted in proportion to the changes in the work rate (time effect: F_(11, 99)_ = 625.470, *p* ˂ 0.001, η^2^ = 0.986) and showed no significant difference between the HCP and placebo groups during the HIIT protocol, as illustrated in Figure 2.

### 3.3. TR-NIRS Profiles of the RF and VL Responses during the HIIT and Recovery

Overall, there were no significant differences between the HCP and placebo groups from rest to recovery in the *RF* and *VL* muscles’ deoxy[Hb + Mb], total[Hb + Mb], and S_t_O_2_ (all interactions: *p >* 0.05, Figure 3).

Table 2 presents the mean peak values of TR-NIRS profiles in the *RF* and *VL* muscles between HCP and placebo across the six peaks (mean peak.1~6) of the HIIT exercise. In the *RF*, the mean peak deoxy[Hb + Mb] and S_t_O_2_ values showed no significant difference between the groups. However, the total[Hb + Mb] was significantly higher with HCP than placebo (HCP: 183 ± 1 vs. Placebo: 181 ± 1 μM, t_(5)_ = 2.973, *p* = 0.031, d = 1.214). In the *VL* muscle, the mean peak values of deoxy[Hb + Mb] displayed a significant difference of approx. 4 μM in deoxy[Hb + Mb] with HCP compared to placebo (HCP: 60 ± 3 μM vs. Placebo: 56 ± 3 μM, t_(5)_ = 3.438, *p* = 0.018, d = 1.404), which demonstrated a possibly greater O_2_ extraction at peaks during the HIIT exercise in the *VL* muscle. However, no significant difference was observed in the *VL* muscle, with similar mean peak values of total[Hb + Mb] and S_t_O_2_ between HCP and placebo groups.

During the subjects’ recovery, in both the *RF* and *VL* muscles, despite the lack of a significant difference between the supplement groups, it was observed that deoxy[Hb + Mb] decreased and S_t_O_2_ increased substantially during recovery compared to the baseline at rest values before the HIIT protocol. In the comparison between Rest and Rec10 in the *RF* muscle, the deoxy[Hb + Mb] following HIIT protocol was significantly lower at 24 ± 9 μM, and its decrease was approx. 30% (time effect: F_(11, 99)_ = 66.131, *p* ˂ 0.001, η^2^ = 0.880, Figure 3A). Similarly, the S_t_O_2_ for both groups was 13 ± 5% and thus significantly promoted at Rec10 (corresponding to a 23% increase from rest) compared to the rest value (time effect: F_(11, 99)_ = 138.472, *p* ˂ 0.001, η^2^ = 0.939, Figure 3C).

The *VL* muscle demonstrated a significantly remarkable decrease in deoxy[Hb + Mb] of 25 ± 7 μM at Rec10, and the percentage of change from rest to final recovery (at Rec.10) was an approx. 53% decrease with deoxy[Hb + Mb] (time effect: F_(11, 99)_ = 76.271, *p* ˂ 0.001, η^2^ = 0.894, Figure 3D). Moreover, the S_t_O_2_ for the *VL* muscle in both groups was 13 ± 4%, which is significantly increased (corresponding to a 17% increase from rest) (time effect: F_(11, 99)_ = 106.156, *p* ˂ 0.001, η^2^ = 0.922, Figure 3F).

### 3.4. Peak Power during HIIT

The peak power (PP) showed a significant difference between the HCP and placebo groups (interaction effect: F_(5, 45)_ = 3.560, *p* = 0.009, η^2^ = 0.283). The PP increased significantly in the first bout with HCP compared to the placebo, 839 ± 112 W vs. 816 ± 108 W, *p* = 0.001. However, no significant difference was observed after that from bouts 2 to 6 (Figure 4).

### 3.5. The Relationship between ΔPCO_2_, ΔHCO_3_^−^ and ΔPP

During the HIIT protocol, a significant increase in the PP in the HCP group compared to the placebo group was observed at the 1st bout. When further examining the relationship between blood gas at rest and the PP at the 1st bout (Figure 5) a significant correlation was noted between the changes in PP at the first bout (ΔPP (HCP-Placebo)), changes in HCO_3_^−^ at rest (ΔHCO_3_^−^ (HCP-Placebo)) (Figure 5A), and changes in PCO_2_ at rest (ΔPCO_2_ (HCP-Placebo)) (Figure 5B). Higher changes in HCO_3_^−^ and PCO_2_ at rest were significantly associated with a greater PP reduction (r = 0.771, *p* = 0.042 and r = 0.809, *p* = 0.027), respectively, and the changes in pH at rest were mainly sensitive to the changes in PCO_2_ (r = 0.907, *p* = 0.004) (Figure 5C).

## 4. Discussion

This research examined the effects of the consumption of a single dose of H_2_ in the form of HCP on healthy subjects’ rest, HIIT performance, and recovery. The results of analyses showed that this dose of HCP significantly increased PCO_2_ and HCO_3_^−^ and showed a slight tendency toward acidosis at rest (i.e., a slightly but not significantly lowered pH). In the HCP group, the increased HCO_3_^−^ was also reflected in the significantly increased BE, and the tendency toward acidosis in pH as a result of the increased PCO_2_ could be additionally augmented as an effect of HCP’s antioxidant properties and the suppression of reactive oxygen species (ROS), as a reduced ventilatory sensitivity to hypercapnia following antioxidant administration has been observed in several studies [7,24,25]. However, further examination of the direct effect of HCP on ROS is needed.

HCP also had no effect on the *RF* and *VL* muscles’ oxygenation status at rest. Comparably, in our previous work [7] when examining the intake of the same dose of HCP (1500 mg) for 3 consecutive days, a slight respiratory acidosis was detected at rest. Reduced rates of ventilation and muscle deoxygenation at rest in the *RF* muscle were also observed. However, the lack of an effect on the skeletal muscle oxygenation status in the *RF* and *VL* muscles in the present study can be attributed to the single dose of HCP and/or possibly the shorter period of ingestion and exposure to changes in PCO_2_ and PO_2_ at rest.

It is possible that the significant increase in PCO_2_ at rest is associated with increased cerebral blood flow (CBF), with hypercapnia resulting in dilatation of the cerebral vasculature as PCO_2_, independently and in conjunction with pH, regulates the CBF [14,15]. Regarding the relationship between CBF and exercise performance, it has been documented that a reduced CBF results in a reduced motor drive to working muscles, consequently reducing the tolerance to whole-body exercise and possibly performance [16,17]. The CO_2_-induced CBF might thus have contributed to the change in the 1st PP performance. However, considering the effect of the HCP supplement on the averaged PP performance up to six exercise repetitions, the PCO_2_-induced increase in CBF with the intake of HCP did not have an ergogenic effect on repeated HIIT, as the PCO_2_ effects on CBF during exercise might be affected by the severity of hypoxia and the intensity and duration of exercise [26,27,28]. In addition, the significantly increased HCO_3_^−^ at rest that was observed in the present study might have further positive effects on the PP by increasing the extracellular buffer capacity, facilitating H^+^ as well as lactate–ion efflux from muscles and delaying muscle fatigue during high-intensity muscle contractions [11,12,13].

As is clear from Figure 5A,B, the relationship between ΔPCO_2_ (HCP-Placebo), ΔHCO_3_^−^ (HCP-Placebo) and ΔPP(HCP-Placebo) is negatively correlated, with HCP improving the PP correlated to changes of approx. 2 mmHg for PCO_2_ and 1.0 mmol∙L^−1^ for HCO_3_^−^. Thus, the HCP supplement contributed to the improvement of the subjects’ HIIT performance through slight increases in PCO_2_ and HCO_3_^−^. Consequently, the HCP supplement resulting in this unique balance among pH, PCO_2_, and HCO_3_^−^ in the present experiment might have exerted a short ergogenic effect on the PP at the first bout at the beginning of the exercise protocol (Figure 4). However, more precise examinations of these factors are necessary to test these conjectures.

With the use of TR-NIRS during HIIT, it was observed that the mean peak values across the six peaks in the *RF* muscle’s total[Hb + Mb] and the *VL* muscle’s deoxy[Hb + Mb] were significantly higher in the HCP group compared to the placebo group. The increased deoxy[Hb + Mb] might suggest impairments in the matching of QO_2_-to-VO_2_ induced by the HCP, resulting in higher fractional O_2_ extractions during the peak power exertion during HIIT in order to meet muscle demands, especially in the *VL* muscle [7]. By contrast, the greater total[Hb + Mb] in the *RF* muscle was associated with an increased microvascular [Hb] volume [29].

Regarding muscle deoxygenation in the both *RF* and *VL* muscles, it was noted that during the subjects’ recovery the deoxy[Hb + Mb] decreased and the S_t_O_2_ increased substantially compared to the baseline at-rest values with the prescribed HIIT protocol, despite the lack of a significant difference between the HCP and placebo supplement groups. In the comparison between Rest and Rec10, the deoxy[Hb + Mb] value following HIIT exercise was significantly lower by 24 ± 9 μM (~30% decrease) in the *RF* and by 25 ± 7 μM (~53% decrease) in the *VL*. Similarly, the S_t_O_2_ for both supplement groups was promoted at 10 min (Rec10) of passive recovery by 13 ± 5% (~23% increase) in the *RF* and 13 ± 4% (~17% increase) in the *VL* (Figure 3). These results demonstrated an abrupt increase in the regional muscle blood flow that was reflected by the tissue oxygen saturation exceeding the resting levels after the HIIT.

As there is a notable difference in the rate of oxygenation among muscle sites during recovery, it has been suggested that differences between the *VL* and *RF* muscles may originate, in part, from higher muscle blood flow [30,31,32] and a greater proportion of more highly oxidative type I fibers in the *VL* muscle [33]. Additionally, judging by the continuing changes observed in the deoxy[Hb + Mb] and S_t_O_2_ values form Rec.1 to Rec.10, it can be assumed that a further increase in the reoxygenation of the *RF* and *VL* muscles would be present after the 10 min of passive recovery in this study.

The first observations regarding muscle deoxygenation kinetics suggested that the HIIT protocol has the advantages of improving the local blood flow and oxygen supply into the working skeletal muscles. Muscle gas exchange is facilitated when the functional cross-sectional area of capillaries is increased, as the vasodilation of small arterioles enhances the functional capillary density (increased number of perfused capillaries), which shortens the diffusion distance for oxygen and other substrates from and to the muscles [34,35]. Moreover, greater local muscle perfusion is associated with enhanced oxidative metabolism, such as fatty acid oxidation [35]. The HIIT protocol employed in the experiment might therefore prove to be a quite advantageous tool in training and improving subsequently exercise performance.

Some study limitations must be considered when interpreting the present results, including the small number of available investigations of hydrogen’s effects prior to and during exercise in healthy subjects. There have been few investigations of the effects and mechanisms of a unique method of delivering H_2_ (such as HCP supplementation), and further research is needed to better comprehend the working conditions and limitations of HCP. In addition, this study did not measure the ventilatory response or provide ROS value changes, and these variables might have provided a better understating of the acute effects of HCP intake during rest and exercise. Finally, as a large individual variation occurs in HCP experiments affecting the acid-base balance, further examinations of the optimal dosage and intake protocols should be considered. However, these limitations do not negate the important findings of this study.

## 5. Conclusions

The results of this study demonstrated that the ingestion of an H_2_-rich calcium powder supplement increased the subjects’ PCO_2_ and HCO_3_^−^ concentrations at rest and improved the peak power in the 1st bout of the HIIT protocol. The HCP had no notable effect on the subjects’ heart rate or muscle O_2_ at rest or recovery; however, the HCP group did exhibit significantly greater O_2_ extraction and microvascular [Hb] volume in their working muscles during the HIIT exercise. The HIIT protocol resulted in improved muscle reoxygenation and may have improved the local muscle perfusion during recovery compared to rest.

These findings suggest that the use of an HCP supplement might produce ergogenic effects on high-intensity exercise and prove advantageous for improving anaerobic HIIT exercise performance.

## Figures and Tables

**Figure 1 nutrients-14-03974-f001:**
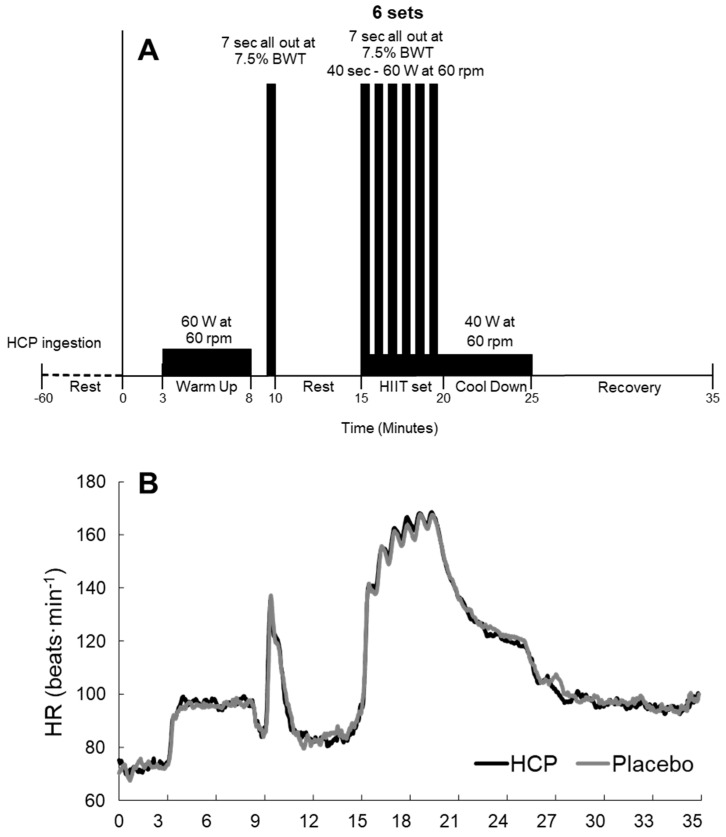
Experimental flow protocol at rest, warm−up, high−intensity intermittent training (HIIT), cool down, and passive recovery (**A**), and the heart rate (HR) response during rest, HITT, and recovery (**B**) in the HCP and placebo groups. HR reflecting the exercise intensity reached values near maximal intensity during the HIIT protocol.

**Figure 2 nutrients-14-03974-f002:**
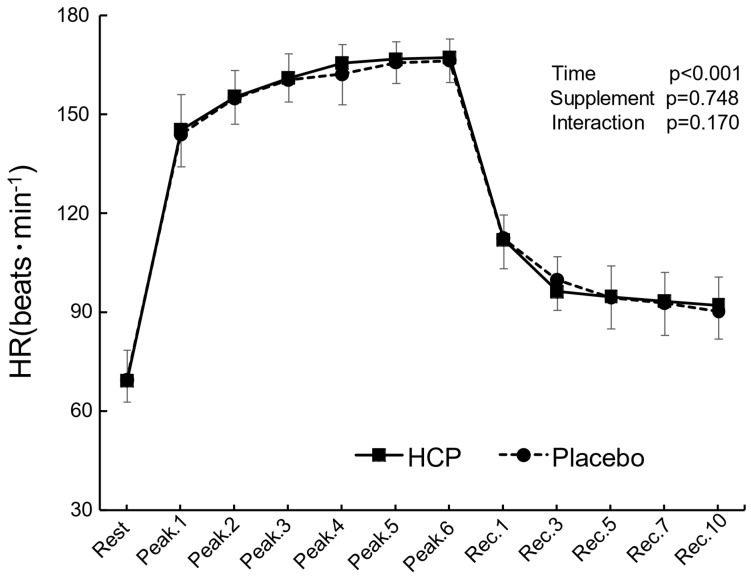
The subjects’ heart rate (HR) response at rest and the peak values during the 7 s all−out bout for each set (Peak.1~6) and the recovery period (Rec.1~10) between the HCP and placebo groups. Error bars: SD. The HR showed no significant difference between the groups.

**Figure 3 nutrients-14-03974-f003:**
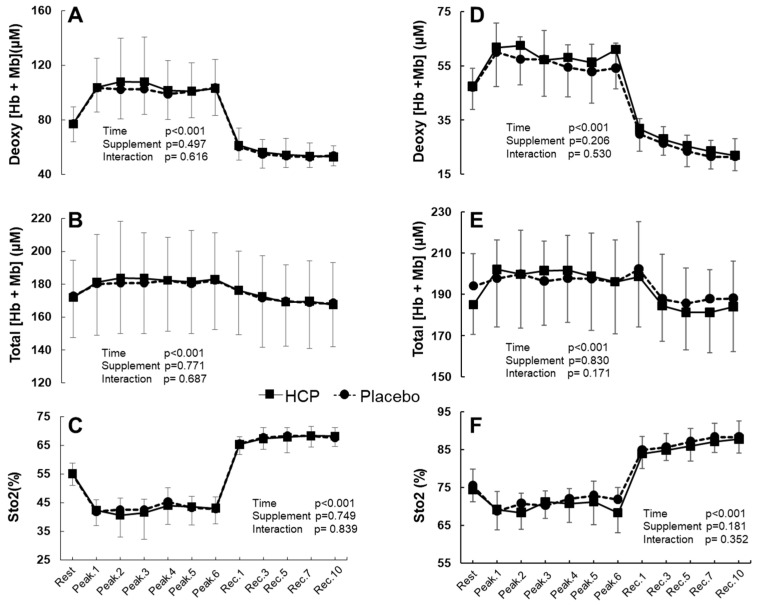
Group mean values for the rectus femoris (*RF*) muscle deoxygenated hemoglobin and myoglobin concentration (deoxy[Hb + Mb]) (**A**), the total hemoglobin and myoglobin concentration (total[Hb + Mb]) (**B**), and tissue O_2_ saturation (S_t_O_2_) (**C**). The vastus lateralis (*VL*) muscle deoxy[Hb + Mb] (**D**), total[Hb + Mb] (**E**), and S_t_O_2_ (**F**) at rest, peak values during the 7 s all-out bout for each set (Peak.1~6), and the recovery period (Rec.1~10) in the HCP and placebo groups. Error bars: SD. Overall values in the *RF* and *VL* muscles showed no significant difference between the groups.

**Figure 4 nutrients-14-03974-f004:**
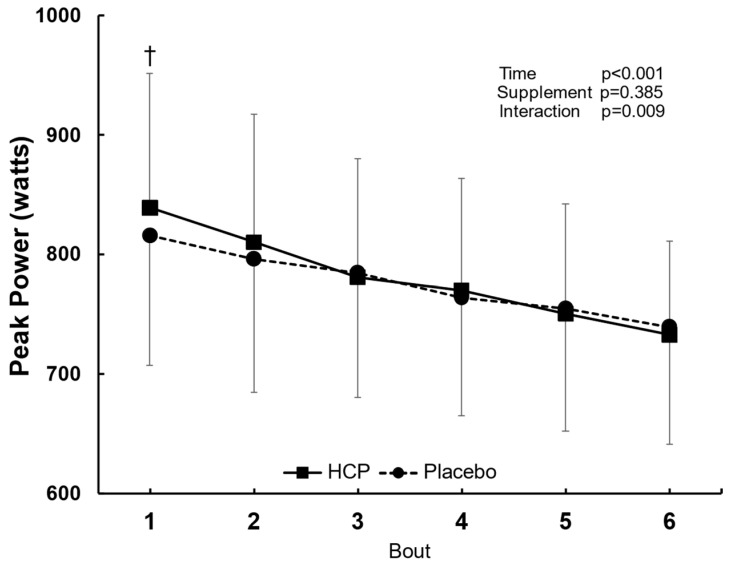
Peak power (PP) output in the 7 s all−out bout during the six repetitions of the HIIT exercise in the HCP and placebo groups. Error bars: SD. ^†^
*p* < 0.01. The PP increased significantly in the 1st bout with HCP compared to the placebo, but no significant difference occurred after that from bouts 2 to 6.

**Figure 5 nutrients-14-03974-f005:**
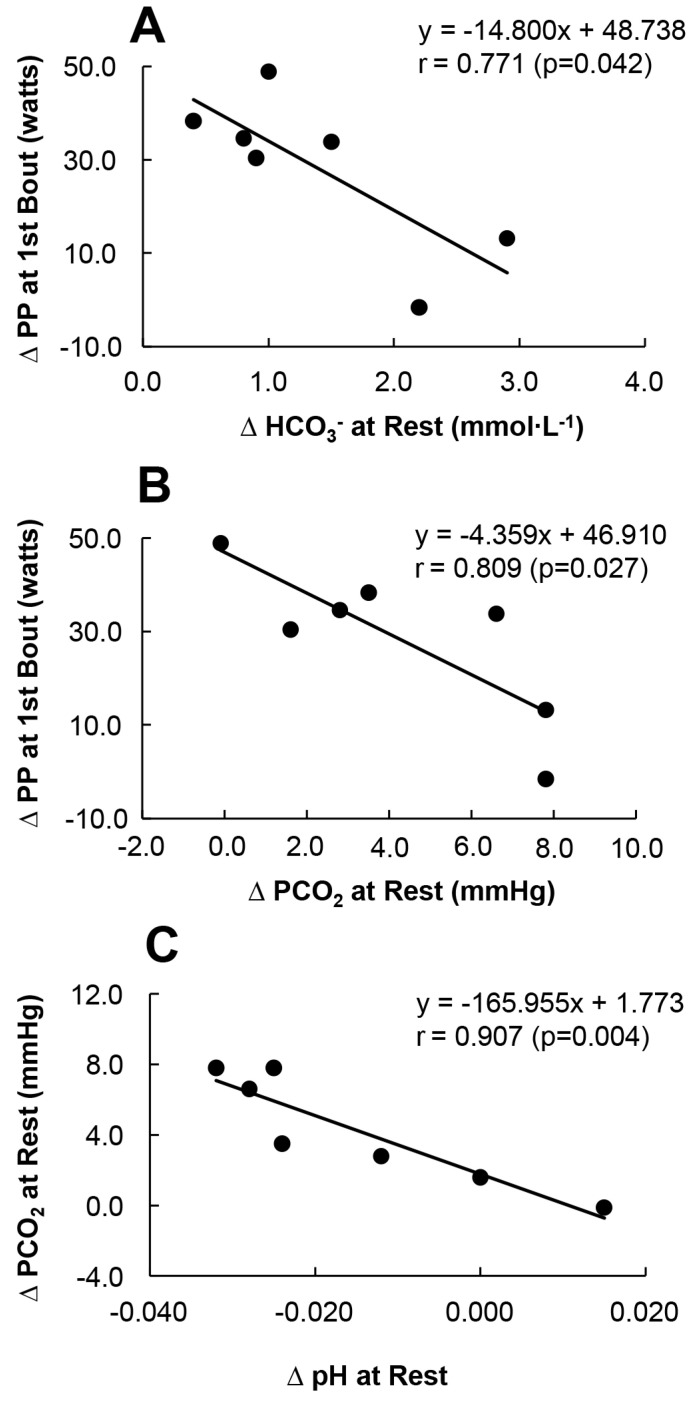
The relationship between changes in the peak power (PP) at the 1st bout (ΔPP (HCP−Placebo)) and changes in the HCO_3_^−^ value at rest (ΔHCO_3_^−^ (HCP−Placebo)) (**A**), the ΔPP at the 1st bout and changes in the PCO_2_ at rest (ΔPCO_2_ (HCP−Placebo)) (**B**), and the ΔPCO_2_ at rest in relation to the changes to pH at rest (ΔpH (HCP−Placebo)) (**C**) in the HCP group. Note that the sample size was 7 subjects. Higher changes in ΔHCO_3_^−^ and ΔPCO_2_ at rest were significantly negatively correlated with ΔPP at the 1st bout (r = 0.771, *p* = 0.042 and r = 0.809, *p* = 0.027, respectively), and the change in pH at rest was sensitive mainly to changes in PCO_2_ (r = 0.907, *p* = 0.004).

**Table 1 nutrients-14-03974-t001:** Mean values of blood gas, electrolytes, metabolic parameters, TR−NIRS and heart rate profiles at rest between HCP and Placebo.

	HCP	Placebo	*p* Value	Effect Size ^a^
**Blood gas**				
pH	7.354 ± 0.021	7.369 ± 0.018	0.059	0.881
PO_2_ (mmHg)	31.3 ± 5.9	41.3 ± 8.7	0.029 *	1.081
PCO_2_ (mmHg)	55.0 ± 4.7	50.7 ± 5.0	0.011 ^†^	1.363
HCO_3_^−^ (mmol∙L^−1^)	30.5 ± 1.6	29.2 ± 1.7	0.006 ^†^	1.571
SO_2_ (%)	54.5 ± 13.4	71.7 ± 12.6	0.022 *	1.158
BE(ecf) (mmol∙L^−1^)	5.0 ± 1.5	3.8 ± 1.4	0.006 ^†^	1.555
BE(b) (mmol∙L^−1^)	3.2 ± 1.1	2.6 ± 0.9	0.034 *	1.034
TCO_2_ (mmol∙L^−1^)	32.3 ± 1.7	30.7 ± 1.8	0.004 ^†^	1.699
Hct (%)	50 ± 4	48 ± 4	0.052	0.915
Hgb (g∙dL^−1^)	17.0 ± 1.4	16.2 ± 1.5	0.033 *	1.038
**Electrolytes**				
Na (mmol∙L^−1^)	141 ± 1	141 ± 1	1.000	0.000
K (mmol∙L^−1^)	4.6 ± 0.4	4.6 ± 0.9	1.000	0.000
Ca (mmol∙L^−1^)	1.30 ± 0.03	1.28 ± 0.05	0.119	0.688
Cl (mmol∙L^−1^)	103 ± 1	103 ± 2	0.078	0.802
AGap (mmol∙L^−1^)	8 ± 1	9 ± 2	0.253	0.477
AGapK (mmol∙L^−1^)	13 ± 1	13 ± 1	0.289	0.439
**Metabolic status**				
Lactate (mmol∙L^−1^)	0.95 ± 0.18	1.03 ± 0.27	0.388	0.351
Glucose (mg∙dL^−1^)	92 ± 6	94 ± 9	0.512	0.264
Creatinine (mg∙dL^−1^)	0.94 ± 0.03	0.93 ± 0.05	0.488	0.279
**TR-NIRS in the RF**				
Total[Hb + Mb] (µM)	172 ± 23	173 ± 25	0.805	0.080
Deoxy[Hb + Mb] (µM)	77 ± 12	77 ± 13	0.913	0.036
StO_2_ (%)	55 ± 4	55 ± 4	0.987	0.005
**TR-NIRS in the VL**				
Total[Hb + Mb] (µM)	185 ± 15	194 ± 16	0.084	0.613
Deoxy[Hb + Mb] (µM)	47 ± 8	47 ± 7	0.925	0.031
StO_2_ (%)	75 ± 3	76 ± 4	0.445	0.253
**Heart Rate** (beats·min^−1^)	69 ± 9	69 ± 7	0.912	0.036

Data are shown as mean ± standard deviation (SD). Significant difference between HCP and Placebo (* *p* ˂ 0.05, ^†^
*p* ˂ 0.01). ^a^ Effect size (Cohen’s d): ≥0.20 small effect, ≥0.50 medium effect, ≥0.80 large effect.

**Table 2 nutrients-14-03974-t002:** Mean values of TR−NIRS profiles in the RF and VL muscles between HCP and Placebo across the 6 peaks of the HIIT protocol.

	HCP	Placebo	*p* Value	Effect Size ^a^
TR-NIRS in the RF				
Total[Hb + Mb] (µM)	183 ± 1	181 ± 1	0.031 *	1.214
Deoxy[Hb + Mb] (µM)	104 ± 3	102 ± 2	0.092	0.851
StO_2_ (%)	43 ± 1	43 ± 1	0.272	0.504
TR-NIRS in the VL				
Total[Hb + Mb] (µM)	200 ± 2	198 ± 1	0.054	1.021
Deoxy[Hb + Mb] (µM)	60 ± 3	56 ± 3	0.018 *	1.404
StO_2_ (%)	70 ± 1	71 ± 1	0.115	0.778

Data are shown as mean ± standard deviation (SD). Significant difference between HCP and Placebo (* *p* ˂ 0.05). ^a^ Effect size (Cohen’s d): ≥0.20 small effect, ≥0.50 medium effect, ≥0.80 large effect.

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
