# Peer review of "The Acute Effects of a Single Dose of Molecular Hydrogen Supplements on Responses to Ergogenic Adjustments during High-Intensity Intermittent Exercise in Humans"

_nutrients, 2022, doi:10.3390/nu14193974_

Round 1

Reviewer 1 Report

This article is well written, organized correctly and is testing research of the effects of single-dose molecular hydrogen (H2) supplements on acid-base status and local muscle deoxygenation during rest, high-intensity intermittent training (HIIT) performance, and recovery. Subjects and Methods are reasonable length, given the premise of the paper. The structure in Results, Study limitations and Conclusion parts are well argued, the logic easy to follow and the benefits of the new testing study are obvious and successful. Figures and tables are very significantly helpful and comprehensible throughout the text. Moreover, the final results are quite optimistic for further use and research even according to the writers.

Although, there is a minor comment that want some attention:

  1. The Introduction part of the article is a little limited about the information that are needed in the course of reading. That part, not only has very little literature references (not overall the article, just the part) but the details about the key components of the experiment are lacking.  More explanatory, maybe some of the information from the results part maybe are more necessary in the introduction.

Author Response

Response to reviewer #1

This article is well written, organized correctly and is testing research of the effects of single-dose molecular hydrogen (H2) supplements on acid-base status and local muscle deoxygenation during rest, high-intensity intermittent training (HIIT) performance, and recovery. Subjects and Methods are reasonable length, given the premise of the paper. The structure in Results, Study limitations and Conclusion parts are well argued, the logic easy to follow and the benefits of the new testing study are obvious and successful. Figures and tables are very significantly helpful and comprehensible throughout the text. Moreover, the final results are quite optimistic for further use and research even according to the writers.

Although, there is a minor comment that want some attention:

Overall response: Thank you for the positive and constructive comments about our manuscript. Addressing these comments have substantially improved the manuscript. We now produce a revised version based on these comments that we hope satisfies the concerns of each of the Reviewers.

The Introduction part of the article is a little limited about the information that are needed in the course of reading. That part, not only has very little literature references (not overall the article, just the part) but the details about the key components of the experiment are lacking.  More explanatory, maybe some of the information from the results part maybe are more necessary in the introduction

Response: Changes were made to increase and expand the information and explanation in the introduction as you suggested. Please refer to the changes from line 64 to line 69. 

Reviewer 2 Report

Thank you for the opportunity to review this work. The manuscript under review assessed whether HCP supplement causes different responses to muscle deoxy[Hb+Mb] and the responses of blood gas values at rest, during anaerobic high-intensity intermittent training (HIIT) exercise, and recovery compared to a placebo supplement.

The study is interesting. However, there are some concerns that need to be addressed before the manuscript could be considered for publication in Nutrients.

Abstract

The abstract is clear and well presented, showing the most relevant information in the manuscript.

Line 25: add W to (839 ± 112).

Introduction

I congratulate the authors on the introduction. It is clear, comprehensive and covers all relevant aspects of the manuscript with a good theoretical framework.

The objectives and hypotheses are well defined and well defined.

Material and methods

Line 71: change the heading to Materials and Methods.

Line 75: it is not necessary to write the units in the text. Therefore, I suggest that the authors present a table (Table 1) to present the characteristics of the subjects.

Line 81: the authors present the study design in this section of Supplements. I suggest that the study design be presented as subsection 2.1 Design and renumber the remaining subsections under materials and methods consecutively.

Line 83: explain the content of the placebo.

Line 104: change 1.4 Exercise test to 2.5 Exercise test.

Therefore, the following subsections of Materials and methods should be renumbered with the suggested considerations.

Line 150: merge this paragraph into the Measurements subsection, deleting the Data analysis subsection in which it was presented.

Results

Lines 171-173: please submit p-values to three decimal places. In case of p<0.05, indicate the exact value to three decimal places.

Table 1. Present it in the table format that is available in the Nutrients template presented. You cannot submit a table in image format.

Line 181: modify the format of the result of F in subscript; F11,99. Likewise, modify the results where this result appears throughout the manuscript.

Discussion

The discussion is well written and referenced, correctly respecting the results, with good internal and external validity.

Lines 256-264: move this paragraph to results. In the Discussion you cannot present results with p-values. Only show in the discussion the presented value judgments of these results (lines 264-266).

Figure 5 should also be moved to results as well as this paragraph.

Line 335: remove the Study Limitations subsection.

Conclusions

Change the heading "Conclusion" to "Conclusions".

Well written, responding to the objectives of the study based on the results presented.

Author Response

Response to reviewer #2

Thank you for the opportunity to review this work. The manuscript under review assessed whether HCP supplement causes different responses to muscle deoxy[Hb+Mb] and the responses of blood gas values at rest, during anaerobic high-intensity intermittent training (HIIT) exercise, and recovery compared to a placebo supplement.

The study is interesting. However, there are some concerns that need to be addressed before the manuscript could be considered for publication in Nutrients.

Overall response: Thank you for the positive and constructive comments about our manuscript. Addressing these comments have substantially improved the manuscript. We now produce a revised version based on the comments that we hope satisfies the concerns of each of the Reviewers.

  1. Abstract:

The abstract is clear and well presented, showing the most relevant information in the manuscript.

Line 25: add W to (839 ± 112).

Response: Thank you kindly, we added measurement unit (W) to line 25.

  1. Material and methods

-Line 71: change the heading to Materials and Methods.

Response: Heading was changed from (Subjects and Methods) to (Materials and Methods).

-Line 75: it is not necessary to write the units in the text. Therefore, I suggest that the authors present a table (Table 1) to present the characteristics of the subjects.

Response: According with your suggestion, we added the description about the subject’s athletic career and 100 m race-record. All subjects are members of college athletic team and had been training for 100 m event for ≥5 years and averaged their 100 m race-record is 11.12 ± 0.38 sec.

-Line 81: the authors present the study design in this section of Supplements. I suggest that the study design be presented as subsection 2.1 Design and renumber the remaining subsections under materials and methods consecutively.

Response: We thank you for your suggestion. The study design description in line 87 corresponds to the method in which the supplement and placebo were introduced in the experiment, so we think it is appropriate to be included in the supplement’s subsection. Additionally, the study description used is quite short, and there is no need in expanding on it further, therefore we feel it is unnecessary to add an additional subsection for it.

-Line 83: explain the content of the placebo.

Response: We described that each capsule contained approx. 375 mg of either HCP or only calcium powder for the placebo in line 91-92.

-Line 104: change 1.4 Exercise test to 2.5 Exercise test. Therefore, the following subsections of Materials and methods should be renumbered with the suggested considerations.

Response: Our apologies for the error. Subsection numbering was corrected (line 111, line 129, line 141, and line 162).

-Line 150: merge this paragraph into the Measurements subsection, deleting the Data analysis subsection in which it was presented.

Response: Subsection header (Data analysis) was removed, and text paragraph was included in the (Measurements) subsection from line 156 to line 161.

  1. Results

-Lines 171-173: please submit p-values to three decimal places. In case of p<0.05, indicate the exact value to three decimal places.

Response: p-values to three decimal places was added for data of PO2, PCO2, and HCO3- from line 175 to line 177.

-Table 1. Present it in the table format that is available in the Nutrients template presented. You cannot submit a table in image format.

Response: Tables 1 and 2 in the template format were newly replaced.

-Line 181: modify the format of the result of F in subscript; F11,99. Likewise, modify the results where this result appears throughout the manuscript.

Response: The result of F was formatted in subscript throughout the manuscript.

  1. Discussion

-The discussion is well written and referenced, correctly respecting the results, with good internal and external validity.

-Lines 256-264: move this paragraph to results. In the Discussion you cannot present results with p-values. Only show in the discussion the presented value judgments of these results (lines 264-266).

-Figure 5 should also be moved to results as well as this paragraph.

Response: According with your suggestion, Lines 256-264 and Figure 5 were moved to the results section from line 244 to line 253.

-Line 335: remove the Study Limitations subsection.

Response: The (Study Limitations) header was removed, and text paragraph was included in the (Discussion) section from line 344 to line 354.

  1. Conclusions

Change the heading "Conclusion" to "Conclusions".

Response: Heading "Conclusion" was changed to "Conclusions".

Reviewer 3 Report

General Comments: I have carefully revised the manuscript entitled “ The Acute Effects of a Single Dose of Molecular Hydrogen Supplements on Responses to Ergogenic Adjustments During High-Intensity Intermittent Exercise in Humans” by Ahad Abdulkarim D. Alharbi and colleagues.
The article is devoted to a topic of substantial importance and matches the range of issues generally covered by the Nutrients.
 The title of the paper corresponds to the issues it addresses. The authors offer a valid overview of the extent of the problem, building on theoretical concepts and research. The literature is recent and relevant. The literature review acknowledges the depth and breadth of investigation in the field
After minor corrections, the article may be approved for publication. In order to increase the value of its content, I suggest the following:

1.  Abstract should have the form "introduction-materials methods-results-conclusion".
2.  Authors should mention the area where population study answered the questions. Where the research was conducted (home, clinic, hospital area etc.)
4. I am not sure whether the study is up to date as no time frame has been specified for it. Please provide this information.
5. Please provide more complete information on project (study) funding.
Paper requires minor revisions, but I commend this paper to the Editor-in-Chief.

Author Response

Response to reviewer #3

General Comments: I have carefully revised the manuscript entitled “The Acute Effects of a Single Dose of Molecular Hydrogen Supplements on Responses to Ergogenic Adjustments During High-Intensity Intermittent Exercise in Humans” by Ahad Abdulkarim D. Alharbi and colleagues.

The article is devoted to a topic of substantial importance and matches the range of issues generally covered by the Nutrients.

 The title of the paper corresponds to the issues it addresses. The authors offer a valid overview of the extent of the problem, building on theoretical concepts and research. The literature is recent and relevant. The literature review acknowledges the depth and breadth of investigation in the field.

Response: Thank you for the positive and constructive comments about our manuscript. Addressing these comments have substantially improved the manuscript. We now produce a revised version based on the comments that we hope satisfies the concerns of each of the Reviewers.

After minor corrections, the article may be approved for publication. In order to increase the value of its content, I suggest the following:

  1. Abstract should have the form "introduction-materials methods-results-conclusion".

Response: We thank you for your kind suggestion. We feel that sections in the abstract can be clearly discerned and differentiated within the text, so the additional sub-headers are unnecessary.

  1.  Authors should mention the area where population study answered the questions. Where the research was conducted (home, clinic, hospital area etc.)

Response: The location where all experimental procedures and testing was conducted in Doshisha University and was additionally added to text in line 99.

  1. I am not sure whether the study is up to date as no time frame has been specified for it. Please provide this information.

Response: Study had been carried out from November 2021 to February 2022.

  1. Please provide more complete information on project (study) funding.

Response: Complete funding information is stated in the funding subsection (line 373-375).
